# Cost–Utility Analysis of Home Mechanical Ventilation in Patients with Amyotrophic Lateral Sclerosis

**DOI:** 10.3390/healthcare9020142

**Published:** 2021-02-01

**Authors:** Ondřej Gajdoš, Martin Rožánek, Gleb Donin, Vojtěch Kamenský

**Affiliations:** Department of Biomedical Technology, Faculty of Biomedical Engineering, Czech Technical University in Prague, nám. Sítná 3105, 272 01 Kladno, Czech Republic; rozanek@fbmi.cvut.cz (M.R.); gleb.donin@fbmi.cvut.cz (G.D.); vojtech.kamensky@fbmi.cvut.cz (V.K.)

**Keywords:** cost–utility analysis, Markov model, home mechanical ventilation, amyotrophic lateral sclerosis

## Abstract

Amyotrophic lateral sclerosis is a disease with rapid progression. The use of mechanical ventilation helps to manage symptoms and delays death. Use in a home environment could reduce costs and increase quality of life. The aim of this study is a cost–utility analysis of home mechanical ventilation in adult patients with amyotrophic lateral sclerosis from the perspective of healthcare payers in the Czech Republic. The study evaluates home mechanical ventilation (HMV) and mechanical ventilation (MV) in a healthcare facility. A Markov model was compiled for evaluation in a timeframe of 10 years. Model parameters were obtained from the literature and opinions of experts from companies dealing with home care and home mechanical ventilation. The cost–utility analysis was carried out at the end of the study and results are presented in incremental cost–utility ratio (ICUR) using quality-adjusted life-years. Uncertainty was assessed by one-way sensitivity analysis and scenario analysis. The cumulative costs of HMV are CZK 1,877,076 and the cumulative costs of the MV are CZK 7,386,629. The cumulative utilities of HMV are 12.57 quality-adjusted life year (QALY) and the cumulative utilities of MV are 11.32 QALY. The ICUR value is CZK-4,403,259. The results of this study suggest that HMV is cost effective.

## 1. Introduction

Amyotrophic lateral sclerosis (ALS) is a condition characterized by degeneration of lower and upper motor neurons resulting in progressive weakness of skeletal muscles including the muscles used for breathing. This condition is rapidly progressive and, on average, within 2 to 4 years from the onset of symptoms, the failure of breathing functions due to respiratory muscle damage becomes a frequent cause of death. Only 5–10% of patients survive more than 10 years. Although there is still no existing treatment for this disease, various stages of the disease are still being studied, so some types of treatment including mechanical ventilation may help to relieve symptoms and thus delay death [1,2,3,4].

In ALS, both non-invasive and invasive mechanical ventilation is being employed. The non-invasive ventilation uses face or nose masks and a volume-cycled or two-cylinder pressure-restricted ventilator to ensure the intermittent overpressure to support ventilation. Tracheostomy ventilation is highly demanding and is associated with risks that put pressure on patients themselves and their caregivers; however, it may prolong survival for many years [3].

The results of foreign studies show that an ideal place for mechanical ventilation (MV) is home care since it lowers costs and increases the quality of life and integration in the community [5]. The cost of maintaining a patient in home care is high but certainly significantly lower than the cost of long-term hospitalization [6].

In the Czech Republic, the transfer of patients, including ALS patients, to home care began in the year 2003. However, experience has shown that the process is very lengthy and inefficient. This is also proved by the fact that there are only about 125 patients with various diagnoses using non-invasive home mechanical ventilation (HMV). In the case of an effective system that can ensure the provision of invasive HMV on both professional and financial levels, we could expect a 5-times higher number of patients using invasive HMV [7]. Currently (from 1 December 2019) there is only one official methodology for providing and financing invasive HMV in the Czech Republic which, however, does not take into account non-invasive HMV. As the foreign guidelines prove [8,9,10,11], it is necessary to address the provision of invasive and non-invasive approaches simultaneously, namely in patients with ALS due to the rapid progress of their condition. The non-invasive HMV may, in the Czech Republic, be covered from public health insurance through payment of the medical device using a voucher [12].

The transfer of a patient to a home environment and the subsequent care is a complex process which should be assessed comprehensively and the decision to provide care in the form of HMV should therefore be based on clear evidence. There are a large number of diagnoses for which it is appropriate to use home mechanical ventilation. Amyotrophic lateral sclerosis was used as it is the most common diagnosis from the neuromuscular group of diseases [13,14,15]. The aim of this study is to perform a cost–utility analysis (CUA) of home mechanical ventilation in adult patients with amyotrophic lateral sclerosis in comparison with mechanical ventilation in a healthcare facility from the perspective of a healthcare payer in the Czech Republic.

## 2. Materials and Methods

To evaluate the cost-effectiveness of HMV from the perspective of a healthcare payer in ALS patients, a state-transition model in the form of a Markov decision tree was created. The model was created on the basis of proposed procedures [8,9,10,11] and consultations with experts participating in HMV treatment in the Czech Republic.

The model consists of two Markov decision trees, each of them representing one method of treatment of ALS patients requiring ventilation support, in both a healthcare facility and a home environment. A state diagram of the Markov model for the home care (“HMV”) stems from 4 states and the state diagram of the Markov model for the care in a healthcare facility (“MV”) stems from 3 states (Figure 1).

Due to the nature of the condition and studies analyzing survival curves, a 10-year timespan was selected for the simulation. Since there are rapid changes in a patient’s condition in the stage of the disease requiring ventilation support, the 10-year timespan was simulated in 120 cycles, i.e., one cycle corresponds to one month.

A Danish study [16] was employed to acquire the probability of transition through the death branch (“Death”), or survival/continuation in the treatment branch (“Survival”), for the health state of non-invasive HMV (“NIV”) and the health state of invasive HMV (“IV”). The probability was obtained by extracting relevant cumulative Kaplan–Meier survival curves concerning the care using non-invasive or invasive mechanical ventilation. To obtain the probability of a patient’s death while on the non-invasive mechanical ventilation (“probability_death_NIV”), a Weibull probability distribution was determined based on the value of the log-likelihood ratio. To obtain the probability of a patient’s death while on the invasive mechanical ventilation (“probability_death_IV”), log-normal probability distribution was determined based on the value of the log-likelihood ratio.

Based on the results of a foreign study [17] observing patients with the non-invasive mechanical ventilation and their transition to the invasive method, the probability of transition from non-invasive to invasive mechanical ventilation (“Transition to IV”) was determined for both Markov trees. The probability of a transition of a patient in a worsening condition while being on the invasive HMV to a hospital (“hospitalization”) and the probability of transition back to a home environment were estimated on the basis of opinions of experts and their internal data on patients. According to the expert opinion (agency ProCare Medical s.r.o.), there are no differences in clinical parameters of a patient in a home environment and a healthcare facility, therefore the Markov tree “HMV” and Markov tree “MV” use the same values for the same health conditions and transitions. Health states and probabilities of mutual transitions are presented in Table 1.

The creation and evaluation of the model was performed through a computer software TreeAge Pro Healthcare [18] and through R program [19].

### 2.1. Cost Identification

The non-invasive HMV costs consist of the HMV technical support costs, nursing care costs and other costs. Bilevel positive airway pressure (BiPAP) devices with a backup respiratory frequency or volume support are used to provide non-invasive HMV. The average price of a BiPAP device is CZK 62,424 and its payment is limited to 7 years. The device is provided with all the accessories such as masks, hoses, humidifiers and filters. The average payment for the masks is CZK 4202, for the hoses CZK 967, for the heated humidifiers CZK 1500 and for the filters CZK 699. Payment due date for accessories is in most cases once a year. The technical support costs were calculated as average costs pursuant to Act No. 48/1997 Coll., on public health insurance [20].

Nursing care costs and other costs (payments for GP visits, payments for medication and other medical materials, rehabilitation care and acquisition costs of medical devices) were analyzed on the basis of data provided by a health insurance company (Zaměstnanecká pojišťovna Škoda) for the year 2019. The obtained data were averaged for the purposes of the model and on the basis of average annual inflation expressed by the increase in an average consumer price index; this amount was recalculated for the year 2020. The overall average costs per patient with the non-invasive HMV, along with estimated costs for the year 2020, are presented in Table 2.

The invasive HMV costs consist of HMV technical support costs, costs of nursing care and other costs. Technical support costs are paid according to one of two payment codes based on a patient´s condition, on the basis of the methodology of general health insurance company CR (VZP) [21]. The nursing care of a patient is ensured by a “certified” provider of HMV and a home care nurse as for the field of expertise (expertise 925). The provided care is governed by a valid payment catalogue of VZP and the relevant decree on setting a one point value for the given year [22]. For the year 2020, the point value was determined by Decree no. 268/2019 Coll., on determination of the point value, the amount of payments of provided services and regulatory restrictions for the year 2020, and it corresponds to the amount of CZK 1.07 [23]. For the purposes of the model, the highest possible use of the nursing care in the length of 180 min a day was taken into account. The analysis of personal costs (payments for GP visits, costs of medication and other medical materials, rehabilitation care and acquisition costs of medical devices) stems from the materials of a university hospital in Brno which calculated these costs for the year 2012. On the basis of average annual inflation rates expressed by the increase in the average consumer price index, this amount was recalculated for the year 2020 [24,25,26]. The overall costs per patient with invasive HMV with estimated costs in the year 2020 are presented in Table 3. The costs are divided according to technical support for mobile and immobile patients.

The mechanical ventilation costs in a healthcare facility include treatment days with a point value for the follow-up intensive care unit (NIP) and follow-up ventilation care (NVP). This medical care is, pursuant to Act No. 372/2011 Coll. on healthcare services and conditions on their provision [27], defined as inpatient care, since it is not possible to provide outpatient care and hospitalization is necessary. The point value of treatment days (ODs) is also assigned on the basis of a category of a medical facility provider, with the overhead costs determined for these treatment days by Decree No. 134/1998 Coll., issuing the list of medical performances with relevant point values [28]. For the year 2020, the point value of overheads for all the mentioned ODs is set at 191.86 points. The point value is determined by Decree No. 268/2019 Coll. on the point value determination of the amount of payments for covered services and regulatory restrictions for the year 2020, corresponding to the amount of CZK 1.18 [29]. NIP may be reported only for 90 days and is followed by the report of NVP. Performances with the point values and payments including overheads for the year 2020 are presented in Table 4.

The model also required the analysis of a patient´s transport costs between a healthcare facility and their home. The data were provided by the emergency medical services of Pilsen region. On the basis of consultations with various experts, a standard patient with non-invasive HMV in a condition demanding intubation and transport to a healthcare facility was first identified for the purposes of transfer costs analysis and modelling. The overall costs consist of medical performances, transport services, separately charged materials or separately charged medication. The other situation describes a patient already with invasive HMV in a condition requiring transport to a healthcare facility. After consulting the experts, this case would not require intubation and some of the medication would not be used, however, overall costs would correspond to the same amount as the patient with non-invasive HMV. The overall costs per patient with invasive and non-invasive HMV, from a healthcare payer perspective, are presented in Table 5.

Each health state is assigned a monthly cost incurred by healthcare payers. Furthermore, the model includes the transitional costs of the healthcare payer for the transport of the patient in case the patient needs to be hospitalized or transported to home care. A 3% discount rate was chosen to adjust future costs [25,30]. The costs used in the care model are listed in Table 6.

### 2.2. Utility Identification

A utility value defining health state “NIV” of the Markov tree “HMV” was derived from a foreign study [31] evaluating quality of life by using the questionnaire SF-36 every 3 months for 1.5 years. However, in order to evaluate the cost-effectiveness analysis, it is necessary to convert the obtained values to one value, namely the index value of the questionnaire EQ-5D. The conversion was performed using an algorithm from a study [32] addressing the conversion of dimensions of the SF-36 questionnaire to the values of the EQ-5D questionnaire. Since the time period was set to 10 years, it was necessary to interpolate the obtained values with a curve, which was again performed in the R program. The average age of patients with ALS (neuromuscular condition) covered by the study was 42.8 years. For the purposes of modelling by setting the lifespan at 120 years, a curve of values from 42.8 to 120 years was interpolated, by which the quality of life would correspond to the value 0.

Rousseau et al. [33], examining the quality of life between non-invasive and invasive methods of ventilation support in patients with ALS, states that the differences in the quality of life are insignificant. The health state “IV”, being “HMV” in the Markov tree, was thus assigned with the same values as the health condition “NIV”. Based on the conclusions of foreign studies [6,34,35,36,37,38,39,40,41,42] proving improvements in the quality of life of patients in home environments, and based on consultations with experts, the utility value for the health states “MV” in the Markov tree and the health state “Hospitalization” decreased by 10%. As with the costs, a 3% rate was selected to discount the benefits [25,30].

### 2.3. Selection of Initial Distribution of Patients

The initial distribution stems from a study [16] providing probabilities of transition through the death branch (“Death”), or survival/continuation in the treatment branch (“Survival”) respectively, for the health conditions of non-invasive (“NIV”) and invasive (“IV”) mechanical ventilation.

### 2.4. Cost–Utility Analysis

The incremental cost–utility ratio (ICUR) calculated according to the following formula was used to present the CUA results:(1)$$ICUR=\frac{\left(Cost_{1}-Cost_{2}\right)}{\left(QALY_{1}-QALY_{2}\right)}=\frac{\Delta Cost}{\Delta QALY},$$

*Cost*_1_ and *QALY*_1_ represent the evaluated intervention and *Cost*_2_ and *QALY*_2_ represent the comparator.

### 2.5. Sensitivity Analysis

A one-way sensitivity analysis was calculated. Due to the low number of probands and lack of further information, it was decided to analyze the sensitivity to the given parameters in the range of 30%. All the input parameters except for the changes in the quality of life values in HMV were gradually changed. A sensitivity analysis in which the discount rate was changed to 0% and 5% was also conducted.

### 2.6. Scenario Analysis

A scenario analysis performs the alteration in the initial distributions entering the model. The entire population was not divided in the individual states but the whole cohort (100%) began in the state of non-invasive (“NIV”) mechanical ventilation.

## 3. Results

### 3.1. Markov Models Analysis

The cohort of patients in the Markov model was distributed within individual cycles depending on the selected transition probabilities in the model. The distribution of the cohort in individual health states within the 10-year timespan is presented in the following figures, for Markov model “HMV” (Figure 2) and for Markov model “MV” (Figure 3).

The distribution of the cohort is also connected with the division of costs and utility. The cumulative costs of the Markov model “HMV” correspond to CZK 1,877,076 after 10 years. The cumulative costs of the Markov model “HV” correspond to CZK 7,386,629 after 10 years. The cumulative costs chart is presented in Figure 4.

When evaluating the Markov models, graphs of cumulative utility arising within the 10-year period of time were also compiled (Figure 5). The cumulative utility in the value of 12.57 quality-adjusted life year (QALY) corresponds to a home environment strategy. The cumulative utility in the value of 11.32 QALY corresponds to a healthcare facility strategy.

The cohort distribution graphs (Figure 2 and Figure 3) show that the distribution is almost identical except for the hospitalization curve, which means that both approaches are very similar, but subsequently it is shown that HMV is much cheaper and provides a higher quality of life without a high risk of rehospitalization.

### 3.2. Cost–Utility Analysis

After evaluating the Markov models, a cost–utility analysis assessing two methods of mechanical ventilation in patients with a diagnosis of ALS from the perspective of a healthcare payer was carried out. In the 10-year timespan within which the care is provided by non-invasive or invasive methods of mechanical ventilation, in both a home environment and a healthcare facility environment, the analysis showed the home environment strategy to be a cost-effective dominant intervention. The CUA results are presented in Table 7.

### 3.3. Sensitivity Analysis

All the input parameters in the range of 30% were changed using a one-way sensitivity analysis. However, the ICUR would be affected in terms of a change in efficiency in the case of a change in a healthcare facility utility assumption. The initial state determines this utility value to be 10% lower than in the home care. Within the sensitivity analysis, the value was varied in the range of 30% from the value allocated to the home care. The sensitivity analysis results are presented in Table 8.

### 3.4. Scenario Analysis

Instead of dividing the entry cohort into individual health states, the entire cohort (100%) entered the health state of a non-invasive method. This scenario did not affect the CUA result, i.e., the ICUR values. There was only a significant reduction in the home mechanical ventilation costs to CZK 755,672 and an increase in mechanical ventilation costs to CZK 7,563,912. There was also a slight increase in QALY, but for both variants.

## 4. Discussion

With its average patient ratio of approximately 1.2/100,000, the Czech Republic is very low in comparison to the given value of 6.6/100,000 population which was published by the European Respiratory Society in the Eurovent study [43]. Since the survey by the European Respiratory Society was carried out, the use of HMV has become more widespread. There has been an increase in the number of patients, better healthcare payment systems have emerged, new indications have expanded and the technology used to provide HMV has improved overall. However, there is a lack of a clear evidence base on the use of HMV and other studies to collect data on the use of HMV in Europe [13]. As a result, and on the basis of available studies, the current prevalence of HMV is expected to be higher than described in the Eurovent study, even in countries where this practice has not been as widespread in the last decade. In addition, the organization lacks comparisons with new publications [14].

For the evaluation of HMV, a model working with health conditions, including Markov models, was selected. In another study, Chandra [44] also used a Markov model to evaluate the effectiveness and cost-effectiveness of individual interventions used in a chronic obstructive pulmonary disease (COPD) patient population. Another study [43], using the findings of a previous study, evaluated cost-effectiveness using a Markov model comparing home mechanical ventilation with the usual care of COPD patients in the United Kingdom.

A Markov model was compiled for the environment of the Czech Republic so that the recommended procedures [8,9,10,11] concerning home mechanical ventilation and care of patients with ALS were followed. In these patients, mechanical ventilation plays a major role in prolonging their lives and it is necessary to start non-invasive mechanical ventilation in time. This is also proven by studies [8,43,45,46,47] examining the correct time of HMV initialization and the use of optimal ventilation techniques. The model ensures the continuity of home mechanical ventilation care, from non-invasive to invasive home mechanical ventilation, which should work in the provision of care for patients with ALS [8,9,10,11].

The results indicate the costs of HMV to be multiple times lower than when providing MV in a healthcare facility. This was already proven by authors in a study examining the results of a long-term mechanical ventilation in patients with ALS [48]. Studies [15,34,49] examining the initial part of home mechanical ventilation also emphasize the cost savings. Initiation during hospitalization is advantageous from the point of view of better patient observation in the adaptation process, but on the other hand there are high costs associated with hospitalization of the patient or even a waiting list. Pallero et al. [50] concluded that outpatient patient adaptation is a more cost-effective strategy for the healthcare system than the adaptation process during hospitalization, as confirmed by the results of a Spanish study [49]. In comparison to non-invasive ventilation, invasive ventilation is more expensive [51]. Bach [52] states that in the environment of the US healthcare system, there is a reduction of costs in home care by about 77%, and confirms again the already mentioned positive benefits of home mechanical ventilation in both non-invasive and invasive forms. In pediatric patients, the benefit is also the contact with parents and family, which again has a positive effect on their development and relationships. The transition also affects the number of intensive care unit hospital beds for other acutely ill patients and further reduces the exposure to nosocomial infections [53].

As proved by foreign studies [6,34,35,36,37,38,39,40,41,42], HMV is associated with the benefits of longer survival and improved quality of life. Assessing health-related quality of life is becoming an increasingly important criterion in research and healthcare, especially in the evaluation of the cost–benefit ratio of medical devices or patients with chronic, incurable disorders [37,41]. Therefore, a cost–utility analysis examining its benefits in terms of quality of life was selected to evaluate HMV and MV variants in patients with ALS. On the basis of the mentioned studies [6,34,35,36,37,38,39,40,41,42] and the opinion of experts, the assumption of a 10% reduction in the quality of life of a patient in a hospital environment was chosen for the cost–utility analysis. The value of the reduction was subsequently varied in the sensitivity analysis and changed the significance of ICUR as the only input, and HMV as the only dominant variant became cost-effective, if we assume that both variants fall below the willingness to pay threshold as they are already covered. Furthermore, a second scenario was analyzed when the distribution of the initial cohort of patients changed. The baseline scenario corresponds to the distribution based on a study [16] used to obtain the probability of survival, in respect to patient deaths in the Markov model. However, the change in the distribution of the input cohort again did not lead to any changes in the results.

The ICUR value, determined on the basis of the payer’s perspective over 10 years of healthcare in patients with ALS, falls into the incremental cost–utility plane quadrant, in which the new intervention has lower costs and a higher effect than the comparator. Thus, this is a cost-effective and dominant intervention. The conclusions of HMV being cheaper and having a greater effect are also proved by the above-mentioned studies with their results of different diagnoses [6,15,34,35,36,37,38,39,40,41,42,48,49,50,52], although none of them directly performed a cost–utility analysis comparing hospital and home use of mechanical ventilation including non-invasive and invasive approaches.

The average survival according to one study [16] used to obtain the survival probability corresponds to 22.9 months without the use of mechanical ventilation, 25.8 months with non-invasive HMV alone, 56.8 months if the initial non-invasive HMV was followed by invasive HMV and 33.8 months if only invasive HMV was used. This study was chosen because of the large number of patients in a European country, a comparison of several approaches and, in particular, because of the monitoring of patients until the end of their lives over a 15-year timeframe. However, other studies, such as the 2016 study on non-invasive HMV or a study [54] on invasive HMV, also provide similar survival results for ALS patients using mechanical ventilation. Bourke [55] addressed various characteristics of patients such as the state of bulbar functions, which, according to the results, affect the quality of life and survival of patients with ALS on MV. These differences were not taken into account in our Markov model. However, it stems from all the results of these studies that life is prolonged with the use of mechanical ventilation, and in addition, the use of HMV increases the quality of life, which can again affect life expectancy. It is possible to further expand the Markov model to the specific part of the therapy before using mechanical ventilation and to deal in more detail with various diseases. With this entry into the Markov model, ALS can be further examined and evaluated in more detail according to progression [56], either between individual types of progression or again from the point of view of the use of home care or healthcare.

The main limitation of this study is the selected perspective of a healthcare payer, as the social system and informal caregivers play a major role in providing HMV. The social system spends a large part of its funding on supporting this care in the form of various social benefits, allowances or pensions. This is furthermore related to the costs of informal care, which is often provided by family members, who may lose their own jobs due to the care provided, incurring informal caregiver costs from a societal perspective. This may be confirmed by the results of a study [57] examining the quality of life of ventilated ALS patients and their caregivers. Job loss occurred with the non-invasive approach of HMV in 19% of cases, and with the invasive approach in up to 60% of cases. According to this study, the average time spent caring for a given patient is 12.6 h for the non-invasive approach and 14.4 h for the invasive approach, which affects other activities of an informal caregiver. Although Mustfa [58] states that a non-invasive approach has no impact on the quality of life of the caregiver and does not place a significant burden on the caregiver, the guidelines [2] state that there is a gradual loss of independence during the illness and assistance with daily activities is needed. This leads to an increase in the burden on informal caregivers and can lead to social, psychological and emotional problems. The quality of life of caregivers caring for patients with HMV is reduced, which is accompanied by physical weakness, and couples may experience reduced sexual activities. Gelinas [59] also reports a significant burden and a significant limitation on other activities.

Based on these limitations, it is appropriate to recommend and conduct further research from a societal perspective, considering all costs associated with HMV and analyzing the quality of life of informal caregivers. However, the already created Markov model can still be used for this. The model can also be subsequently used for other diagnoses or another country, by altering the input data such as transition probabilities, costs and benefits. This study could also be suitable for assessing the situation where certain areas do not have good accessibility in terms of distance to medical facilities. It is therefore also possible to evaluate other diseases typical of HMV and different perspectives according to the created model. Currently, in connection with COVID-19, it would be possible to consider the benefit of HMV, in which in the longer term there would be a release of beds in hospital facilities for acute patients with COVID-19 using mechanical ventilation.

## 5. Conclusions

The results of the cost–utility analysis show that in the given setting of the study we can consider HMV as cost effective. Subsequent results of the sensitivity analysis and scenario analysis do not show significant changes in the results. Nevertheless, it is recommended to continue the research from a societal perspective and to use the created model for the research of typical HMV diseases.

## Figures and Tables

**Figure 1 healthcare-09-00142-f001:**
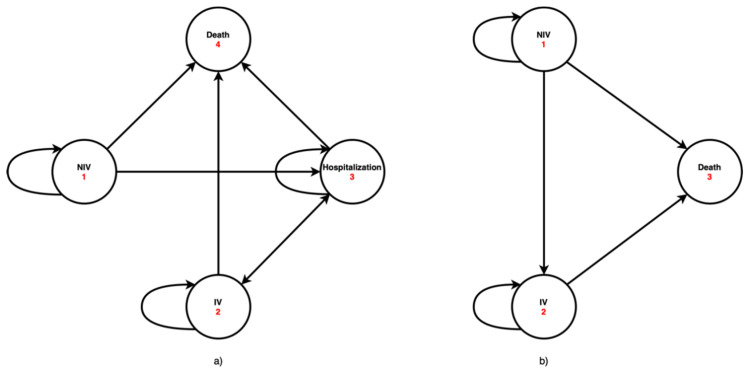
State diagram of the Markov model (**a**) home mechanical ventilation, (**b**) mechanical ventilation.

**Figure 2 healthcare-09-00142-f002:**
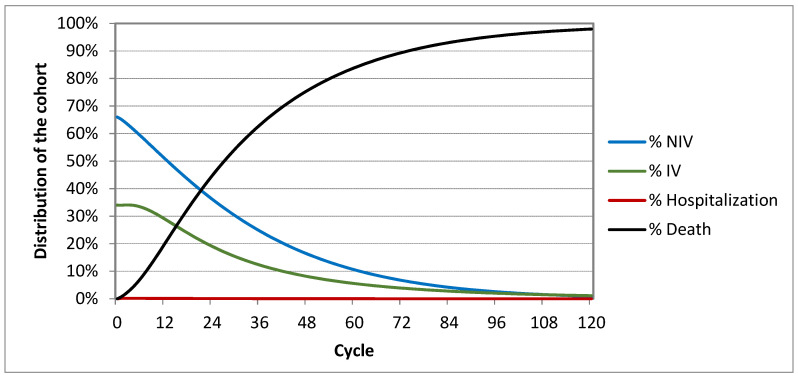
Distribution of cohort in individual cycles of home mechanical ventilation (“HMV”).

**Figure 3 healthcare-09-00142-f003:**
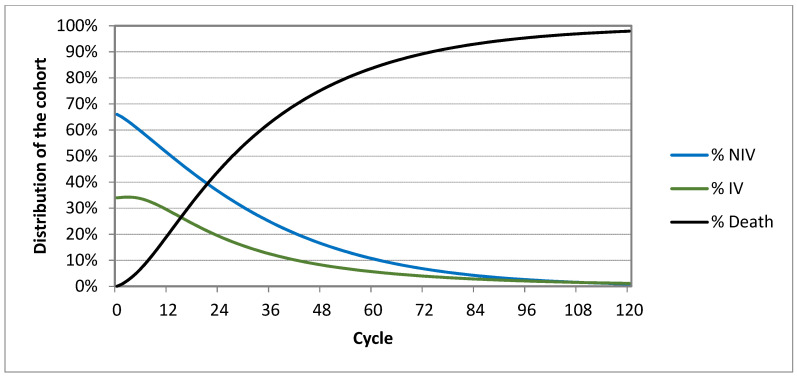
Distribution of cohort in individual cycles of mechanical ventilation (“MV”).

**Figure 4 healthcare-09-00142-f004:**
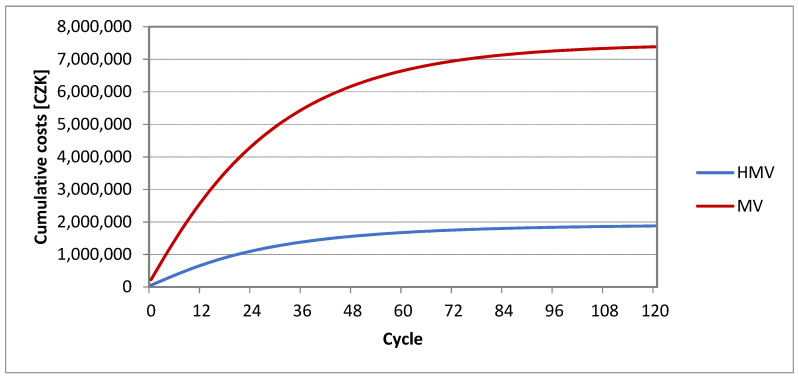
Cumulative costs.

**Figure 5 healthcare-09-00142-f005:**
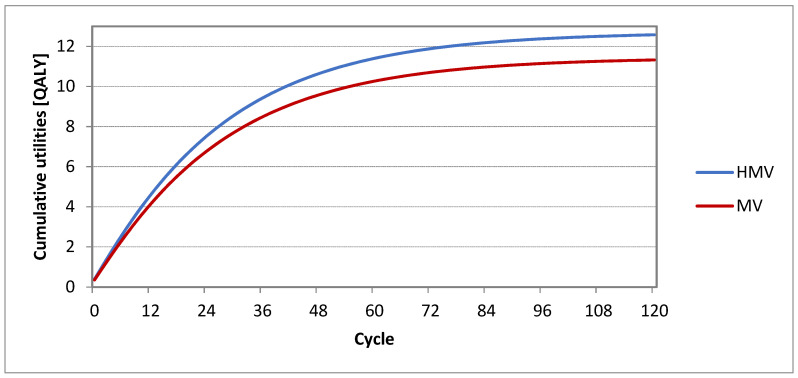
Cumulative utilities.

**Table 1 healthcare-09-00142-t001:** Health states and probabilities of mutual transitions.

**Health State HMV**	**Branch**	**Probability**	**Data Source**	**Health State**	**Probability**	**Data Source**
**NIV**	Survival	Evaluation to the sum of 1	--	NIV	0.99812	[17]
				Hospitalization	0.00188	[17]
	Death	Weibull probability	[16]	Death	--	--
**IV**	Survival	Evaluation to the sum of 1	--	IV	0.999	Expert estimation ^1^
				Hospitalization	0.001	Expert estimation ^1^
	Death	Log-normal probability	[16]	Death	--	--
**Hospitalization**	--	--	--	Hospitalization	0.01	Expert estimation ^1^
	--	--	--	IV	0.95	Expert estimation ^1^
	--	--	--	Death	0.04	Expert estimation ^1^
**Death**	--	--	--	--	--	--
**Health state MV**	**Branch**	**Probability**	**Data source**	**Health state**	**Probability**	**Data source**
**NIV**	Survival	Evaluation to the sum of 1	--	NIV	0.99812	[17]
				IV	0.00188	[17]
	Death	Weibull probability	[16]	Death	--	--
**IV**	--	--	--	IV	Evaluation to the sum of 1	[16]
	--	--	--	Death	Log-normal probability	[16]
**Death**	--	--	--	---	--	--

^1^ Experts from company dealing with home care and home mechanical ventilation in the Czech Republic.

**Table 2 healthcare-09-00142-t002:** The overall average costs per patient with the non-invasive home mechanical ventilation for the year 2020.

Type of Costs	1 Day	1 Month (30 Days)	1 Year (365 Days)
Technical support	CZK45	CZK 1338	CZK 16,286
Nursing care and other costs	CZK 524	CZK 15,720	CZK 191,266
Total	CZK 569	CZK 17,058	CZK 207,552

**Table 3 healthcare-09-00142-t003:** The overall costs per patient with invasive home mechanical ventilation for the year 2020.

	Mobile Patient		Immobile Patient	
Type of Costs	1 Month (30 Days)	1 Year (365 Days)	1 Month (30 Days)	1 Year (365 Days)
Technical support	CZK 20,550	CZK 250,025	CZK 23,550	CZK 286,525
Nursing care	CZK 38,809	CZK 472,175	CZK 38,809	CZK 472,175
Other costs	CZK 68,192	CZK 829,671	CZK 68,192	CZK 829,671
Total	CZK 127,551	CZK 1,552,006	CZK 130,551	CZK 1,588,371

**Table 4 healthcare-09-00142-t004:** Payments for treatment days of long-term nursing care including the value of overheads for the year 2020.

Performance Code	Medical Performance	Point Value	Payment for 1 Day
OD 00017	Follow-up intensive care	9364	CZK 11,276
OD 00015	Follow-up ventilation care	6150	CZK 7483

**Table 5 healthcare-09-00142-t005:** The overall transport costs.

Type of Costs	Costs
Medical performance	CZK 2055.46
Transport services	CZK 2725.68
Separately charged material or medication	CZK 450
Total	CZK 5231.14

**Table 6 healthcare-09-00142-t006:** The costs used in the care model.

**Health State HMV**	**Costs Per Month (30 Days)**
**NIV**	CZK 17,058
**IV**	CZK 129,051
**Hospitalization**	1st to 3rd month CZK 333,277 (NIP), following months CZK 224,502 (NVP)
**Transfer**	CZK 5231.14
**Death**	--
**Health state MV**	**Costs per month (30 days)**
**NIV**	CZK 224,502 (NVP ^1^)
**IV**	CZK 224,502 (NVP ^1^)
**Death**	--

^1^ For the purposes of Markov “MV” model, only the treatment day NVP is reported.

**Table 7 healthcare-09-00142-t007:** Cost–utility analysis.

Strategy	Costs [CZK]	Incremental Costs [CZK]	Utilities [QALY]	Incremental Utilities [QALY]	C/QALY [CZK]	ICUR [CZK]
HMV	1,877,076	0	12.57	0	149,292	0
MV	7,386,629	−5,509,554	11.32	1.25	652,418	−4,403,259

**Table 8 healthcare-09-00142-t008:** Sensitivity analysis of utility in the healthcare facility.

	Strategy	Costs [CZK]	Utilities [QALY]	ICUR [CZK]
−30%	HMV	1,877,076	12.57	0
	MV	7,386,629	8.81	−1,463,995
−20%	HMV	1,877,076	12.57	0
	MV	7,386,629	10.06	−2,197,399
−10%	HMV	1,877,076	12.57	0
	MV	7,386,629	11.32	−4,403,259
0%	HMV	1,877,076	12.58	0
	MV	7,386,629	12.58	1,143,504,522 ^1^
+10%	HMV	1,877,076	12.58	0
	MV	7,386,629	13.84	4,369,607
+20%	HMV	1,877,076	12.58	0
	MV	7,386,629	15.10	2,188,986
+30%	HMV	1,877,076	12.58	0
	MV	7,386,629	16.35	1,460,256

^1^ diff. < 0.01.

## Data Availability

Data sharing not applicable.

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
