# Peer review of "Cost–Utility Analysis of Home Mechanical Ventilation in Patients with Amyotrophic Lateral Sclerosis"

_healthcare, 2021, doi:10.3390/healthcare9020142_

Round 1

Reviewer 1 Report

This is limited to Amyotrophic lateral sclerosis.

However, home mechanical ventilation (HMV) is an issue that is sufficiently important for the Corona19 situation.

It is a study that will be used as an important material for patient care in the most untact era.

Although there is a slightly cost-effective analysis, it can be used as an efficient basis for application to isolation patients.

Reviewer 2 Report

The aim of the study was to perform a cost-utility analysis of home mechanical ventilation (HMV) in adult patients with amyotrophic lateral sclerosis (ALS) in comparison with mechanical ventilation (MV) in a healthcare facility from the perspective of a healthcare payer in the Czech Republic. To evaluate the cost-effectiveness of HMV from the perspective of a health care payer in the ALS patients, the State-Transition Model in the form of a Markov decision tree was created on the basis of proposed procedures and consultations with experts participating in HMV treatment in the Czech Republic.
A Markov process is a stochastic process for which predictions can be made regarding future outcomes based solely on its present state and — most important — that such predictions are just as good as the ones made knowing the process's full history. In other words, conditional on the present state of the system, its future and past states are independent. In this paper, the authors used the technique very much like economists who analyze cost-benefit ratios applied to health care models designed by health care organizations, medical centers and governments, based on available resources.
Their results showed that the costs of HMV to be about several (5-7) times lower than when providing MV in a healthcare facility, findings previously demonstrated by these authors and others. Thus the findings are original primarily as they apply to the health care system in the Czech Republic. Their findings are probably applicable to other countries with differing resources.

Specific comments:
1. The language is somewhat dense and could use some streamlining and editorializing. Many sentences can be shortened to ease comprehension. Repetition of some statements in the introduction and discussion can also be eliminated to make reading easier.

2. The main criticism I have is that the authors define ALS as a rapidly progressive condition in all patients and assume that the analysis they applied can be applied to all patients who develop respiratory failure from ALS. Progression of respiratory muscle weakness in ALS varies considerably among patients and could represent distinct disease phenotypes. Ackrivo et al (Am J Respir Crit Care Med 2019; 195(1) https://doi.org/10.1164/rccm.201604-0848OC, PubMed: 27494149 ) analyzed a single-center cohort of 837 patients with ALS and reported three kinds of FVC decline over time, termed “stable low,” “rapid progressor,” and “slow progressor.” The present authors assumed a 10-year time scope selected for the simulation. Given that patients will have differing rates of respiratory decline, the time of onset of respiratory failure as well as time spent on assisted ventilation until death will vary from patient to patient. Therefore, the authors would have results more reflective of these phenotypes if the analysis were done in patients with different severities and rates of progression of respiratory failure. In other words, I do not think it is appropriate to lump all ALS patient in one basket.

3. The authors also employed cumulative Kaplan-Meier survival curves while patients were on non invasive or invasive mechanical ventilation. In addition, one would assume that HMV technical support costs, nursing care costs and
other costs would vary considerably, depending on the phenotype of ALS progression. The Weibull probability distribution would also change based on the rate of decline in respiratory function.
I think the paper can be improved by applying the Markov model to these different rates of progression rather than assuming a single rate.

Reviewer 3 Report

  • It would be useful to better describe the patient cohort, see for example 10.1016/j.nicl.2018.08.001
  • Except for the hospitalization line, Figure 2 and figure 3 are the same (the curves assume the same values). I don't understand the curves in what they differ.
  • There are some typo (es. Title 3.3)
  • Two important factors are not taken into account when evaluating costs:
  1. The training of the people who manage patients in the hospital
  2. The role of the caregiver (even if included as a limitation of the study).

The absence of these two factors makes the job much weaker.
